# Usnic Acid Derivatives Inhibit DNA Repair Enzymes Tyrosyl-DNA Phosphodiesterases 1 and 2 and Act as Potential Anticancer Agents

**DOI:** 10.3390/genes14101931

**Published:** 2023-10-12

**Authors:** Alexandra L. Zakharenko, Nadezhda S. Dyrkheeva, Olga A. Luzina, Aleksandr S. Filimonov, Evgenii S. Mozhaitsev, Anastasia A. Malakhova, Sergey P. Medvedev, Suren M. Zakian, Nariman F. Salakhutdinov, Olga I. Lavrik

**Affiliations:** 1Institute of Chemical Biology and Fundamental Medicine, Siberian Branch of the Russian Academy of Sciences, 8 Akademika Lavrentieva Ave., 630090 Novosibirsk, Russia; dyrkheeva.n.s@gmail.com (N.S.D.); lavrik@niboch.nsc.ru (O.I.L.); 2N. N. Vorozhtsov Novosibirsk Institute of Organic Chemistry, Siberian Branch of the Russian Academy of Sciences, 9 Akademika Lavrentieva Ave., 630090 Novosibirsk, Russia; luzina@nioch.nsc.ru (O.A.L.); alfil@nioch.nsc.ru (A.S.F.); mozh@nioch.nsc.ru (E.S.M.); anvar@nioch.nsc.ru (N.F.S.); 3Federal Research Center Institute of Cytology and Genetics, Siberian Branch of the Russian Academy of Sciences, 10 Akademika Lavrentieva Ave., 630090 Novosibirsk, Russia; amal@bionet.nsc.ru (A.A.M.); medvedev@bionet.nsc.ru (S.P.M.); zakian@bionet.nsc.ru (S.M.Z.)

**Keywords:** tyrosyl-DNA phosphodiesterase 1 (Tdp1), tyrosyl-DNA phosphodiesterase 2 (Tdp2), usnic acid, Tdp2 inhibitors, Tdp1 inhibitors

## Abstract

Tyrosyl-DNA phosphodiesterase 1 and 2 (Tdp1 and Tdp2) are DNA repair enzymes that repair DNA damage caused by various agents, including anticancer drugs. Thus, these enzymes resist anticancer therapy and could be the reason for resistance to such widely used drugs such as topotecan and etoposide. In the present work, we found compounds capable of inhibiting both enzymes among derivatives of (−)-usnic acid. Both (+)- and (−)-enantiomers of compounds act equally effectively against Tdp1 with IC_50_ values in the range of 0.02–0.2 μM; only (−)-enantiomers inhibited Tdp2 with IC_50_ values in the range of 6–9 μM. Surprisingly, the compounds protect HEK293FT wild type cells from the cytotoxic effect of etoposide (CC_50_ 3.0–3.9 μM in the presence of compounds and 2.4 μM the presence of DMSO) but potentiate it against Tdp2 knockout cells (CC_50_ 1.2–1.6 μM in the presence of compounds against 2.3 μM in the presence of DMSO). We assume that the sensitizing effect of the compounds in the absence of Tdp2 is associated with the effective inhibition of Tdp1, which could take over the functions of Tdp2.

## 1. Introduction

Radiation therapy and some of the traditional anti-cancer chemotherapeutic agents exert their anti-cancer effect through DNA damage. The development of inhibitors of DNA repair enzymes and factors is a promising area of pharmacology and is one of the ways to create effective cancer therapy, especially to combat drug-resistant tumors. The inhibition of key DNA repair enzymes in model systems significantly enhances the effectiveness of traditional methods of treatment [1,2].

Topoisomerase inhibitors such as topotecan, irinotecan, etoposide, and doxorubicin are commonly used anticancer drugs [3,4]. To control DNA topology, topoisomerases 1 and 2 (Top1 and Top2) generate breaks in the DNA backbone by forming phosphotyrosyl bonds between the DNA end and the topoisomerase catalytic tyrosine residue during the formation of transient Top-DNA cleavage complexes (TOPcc). These transient complexes can be stabilized with the drugs listed above, resulting in disruption of genomic stability and cell death if such “stuck” complexes are not repaired [5]. The repair of such damage is carried out by tyrosyl-DNA phosphodiesterases 1 and 2 (Tdp 1/2), which eliminate blocking damage from the 3′-(Top1) and 5′-ends (Top2) of DNA [6].

Tdp1 belongs to the phospholipase superfamily that hydrolyzes the DNA phosphodiester bond [7]. Tdp1 removes DNA damages created by Top1, its inhibitor camptothecin, and other anticancer drugs. Tdp1 cleaves the 3′ phosphodiester bond between covalent adducts of different origins and the 3′ end of DNA [8,9,10,11]. Such adducts are formed, among other ways, under the action of radiotherapy and anticancer drugs [10,11]. Such drugs are the anticancer drugs topotecan and irinotecan, which fix short-lived TOPcc and, thereby, lead to cell death. Thus, by eliminating these complexes, Tdp1 reduces the effectiveness of therapy. The idea that Tdp1 is responsible for resistance to these drugs is supported by a large number of studies, both in vitro and in vivo. Cells or organisms deficient in Tdp1 activity are more sensitive to the effects of Top1 poisons [12,13,14,15,16]. On the other hand, tumor cells with overexpression of Tdp1 are less sensitive to these drugs [17,18,19,20,21]. Also, it was shown that overexpression of Tdp1 is associated with chromosomal instability and is observed in some types of cancer, for review see [22].

The human DNA repair enzyme Tdp2 belongs to the family of metal-dependent phosphodiesterases and hydrolyzes 5′-phosphotyrosine, the product of the covalent attachment of Top2 to DNA [23,24]. Top2 inhibitors (etoposide and doxorubicin) stabilize the Top2-DNA covalent complex and induce cell death [25]. Top2 inhibitors today are clinically used as anticancer drugs. Tdp2 activity could reduce the efficiency of these drugs and vice versa, the deficiency of Tdp2 leads to an increase in sensitivity to Top2 inhibitors [23,26]. Thus, Tdp2 inhibitors can increase the effectiveness of chemotherapy through synergy with Top2 inhibitors. Such sensitization of the antitumor effect was successfully implemented for inhibitors of another similar pair of enzymes, Tdp1/Top1 [22,27,28]. Top1, like Top2, forms a covalent complex with DNA but from the 3′-end.

Of note is the recently discovered ability of Tdp1 and Tdp2 to take over each other’s functions, albeit with lower efficiency [29,30,31]. This makes either the combined use of selective inhibitors of these two enzymes highly promising or the creation of agents capable of simultaneously inhibiting Tdp1 and Tdp2.

The list of Tdp1 inhibitors is quite extensive and includes both natural substances and their derivatives (detailed in the review [22]) and synthetic compounds [27] (Figure 1). In the series of our works, we found effective Tdp1 inhibitors including those that are derivatives of usnic acid [22]. IC_50_ values for compounds **1**–**4** (Figure 1) and others in our works were measured on purified recombinant human Tdp1 by real-time fluorescence measurement. We have shown that the application of Tdp1 inhibitors (compounds **1**–**4**, Figure 1) in combination with topotecan increases its antitumor and antimetastatic efficacy against tumors in mice in vivo [32,33,34,35]. The usnic acid derivatives are the most effective Tdp1 inhibitors known to date (similar to compound **4** in Figure 1), acting in the nanomolar concentration range and capable of potentiating the antitumor and antimetastatic effects of topotecan in vitro and in vivo [22].

The list of known Tdp2 inhibitors is significantly limited, mainly represented by deazaflavins [36,37] and quinoline derivatives [38,39] (for example compounds **5** and **6**, Figure 2). Previously, our team found several mild Tdp2 inhibitors among derivatives of usnic acid and deoxycholic acid (compounds **7** and **8**, Figure 2) [40,41]. IC_50_ values in these works were measured on purified recombinant human Tdp2 by separation of reaction products in polyacrylamide gel.

Usnic acid is a natural compound of the dibenzofuran class, a major metabolite of a wide variety of lichens. This metabolite exists in two enantiomeric forms, the biological activity of which can significantly differ. As a rule, only one of the enantiomers is contained in various types of lichens. The content of (+)-usnic acid is quite high, which, together with the simple procedure of isolation and high optical purity, makes it commercially available. The availability of (+)-usnic acid, its structure, and the presence of functional groups available for chemical modification makes it an attractive object for the synthesis of various derivatives, often having a more pronounced biological activity than the native compound itself. (−)-Usnic acid is less common and less commercially available. The biological activity of usnic acid derivatives can also vary depending on the structure of the chiral centers of both the usnic acid core and substituents [42,43].

The aim of this work was to study the inhibitory activity of usnic acid derivatives, which are effective inhibitors of Tdp1 against Tdp2 in order to identify possible dual Tdp1/Tdp2 inhibitors. The inhibition of both enzymes should lead to an increase in the efficiency of sensitization and also to expand the boundaries of the use of sensitizers in anticancer therapy, including therapy with both Top1 poisons (camptothecin derivatives) and with Top2 inhibitors (etoposide and doxorubicin).

In this work, we have shown that some highly effective inhibitors of Tdp1 that we have previously discovered [44] are also capable of inhibiting Tdp2. If Tdp1 was inhibited with both usnic acid enantiomers with approximately the same efficiency, then only (−)-enantiomers inhibited Tdp2.

## 2. Materials and Methods

### 2.1. Gel-Based Tdp2 Activity Assay

Modified oligonucleotide with tyrosine (5′-tyrosine-AAC GTC AGG GTC TTC C-FAM-3′) for the detection of Tdp2 activity in polyacrylamide gel was synthesized in the Laboratory of Nucleic Acid Chemistry at the ICBFM (Novosibirsk, Russia). Recombinant human Tdp2 was expressed and isolated as described in [40]. Tdp2 gel-based assays are also described there [40]. Briefly, 20 µL samples contained 100 nM substrate and 200 nM Tdp2 in the absence or presence of an inhibitor in a buffer containing 50 mM Tris-HCl, pH 8.0, 7 mM β-mercaptoethanol, 50 mM NaCl, 10 mM MgCl_2_, and 0.05 mg/mL BSA. We incubated the samples for 10 min at 37 °C and then reactions were terminated by the addition of a buffer for gel loading with TBE, 0.1% xylene cyanol, 0.1% bromophenol blue, 7 M carbamide, and 20 mM EDTA and heated at 90 °C for 5 min. The reaction products were analyzed by electrophoresis in a 20% denaturing PAGE with 7 M carbamide (acrylamide to bisacrylamide ratio 19:1) and by gel scanning on Typhoon FLA 9500 phosphorimager (GE Healthcare, Boston, MA, USA). Then, the data were calculated with QuantityOne 4.6.7 software (Bio-Rad Laboratories, Inc., Hercules, CA, USA).

### 2.2. Cytotoxicity Assay

The intrinsic cytotoxicity of the compounds was examined against cell line HEK293FT (human embryonic kidney) with a standard MTT test [45]. The cells were grown in DMEM-F12 medium, with 50 IU/mL penicillin and 50 µg/mL streptomycin (Thermo Fisher Scientific, Waltham, MA, USA) in the presence of 10% fetal bovine serum (Biolot, Saint-Petersburg, Russia) in 5% CO_2_ atmosphere. Tested compounds were added after formation of a 30–50% monolayer to the medium (1:100 added reagents to the total volume of the culture medium, thus the final concentration of DMSO was 1%) and the cell culture was monitored for 3 days. Control cells were grown in the presence of 1% DMSO. The measurements of cytotoxicity were carried out in three repeats.

Different concentrations of etoposide and fixed concentrations of usnic acid derivatives 10 μM were used to evaluate the effect of the tested compounds on the cytotoxicity of etoposide.

### 2.3. Tdp2 Knockout HEK293FT Cell Clones

We performed a selection of protospacers for human *Tdp2* gene knockout with the Benchling CRISPR tool (https://www.benchling.com/) (accessed on 13 February 2020). Two 20-nt protospacer sequences were selected for deletion of exon 6 (PAM sequences in brackets) of the human Tdp2 gene (NM_016614.3): Tdp2-gRNA1 ATGCCTTTGGTGTATTGAGG(AGG) and Tdp2-gRNA2 CAGAAATTAGCCGGGCATTG(TGG). The scheme of protospacers in the *Tdp2* gene is shown on Appendix A. Oligonucleotides were cloned in a pSpCas9(BB)-2A-GFP (PX458) plasmid that was a kind gift from Dr. Feng Zhang (Addgene plasmid #48138; http://n2t.net/addgene:48138 (accessed on 13 February 2020), RRID:Addgene_48138) as previously described [46]. Transfection-grade DNA plasmids were isolated with the Plasmid Plus Midi Kit (QIAGEN, Hilden, Germany).

We transfected HEK293FT cells with the constructed plasmids (0.25 µg of each) pX458-Tdp2-gRNA1 and pX458-Tdp2-gRNA2 using Lipofectamine 3000 Reagent (Thermofisher Scientific, Waltham, MA, USA). We used growth medium containing DMEM/F12 (Thermofisher Scientific, MA, USA) 1:1, 10% fetal bovine serum (Thermofisher Scientific, MA, USA), 100 U/mL penicillin–streptomycin (Thermofisher Scientific, MA, USA), and GlutaMAX (1×) (Thermofisher Scientific, Waltham, MA, USA). Next, (nearly 48 h after transfection), transfected GFP-positive cells were plated onto a 96-well plate and one cell per well by cell sorting using BD FACSAria III Cell Sorter (BD Biosciences, East Rutherford, NJ, USA) was performed. Single-cell clones grew for nearly two weeks and then they were replicated to another 96-well plate. Thus, we had two equal 96-well plates and used one plate for PCR analysis of the deletion in the *Tdp2* gene while the second plate was used for the mutant cell clone growth.

Genome DNA was analyzed by PCR amplification of the target region with primers to detect the presence of the CRISPR/Cas9-mediated deletions (Tdp2-Del-F 5′-GCCAGGCCCTTAATAATACAGC-3′ and Tdp2-Del-R 5′-TGTTTGCTCATTCACACTCCAG-3′) and wild-type alleles (Tdp2-In-F 5′-CGTGTCAGGAAATGAGCTTTG-3′ and Tdp2-In-R 5′-GGAACAATTATCACGGTCCAATC-3′). The scheme of primer location in the Tdp2 gene is shown in Appendix A. The reactions were run using BioMaster HS-Taq PCR-Color (2×) (Biolabmix, Novosibirsk, Russia) on an S1000 Thermal Cycler (Bio-Rad, Singapore) with the program as follows: 95 °C for 3 min; 35 cycles: 95 °C for 30 s; 65 °C for 30 s; 72 °C for 30 s; and 72 °C for 3 min. The reaction products were resolved in 1.5% agarose gel containing ethidium bromide. As a result, we found two HEK293FT cell clones which contained only alleles with deletions (Appendix A).

Homozygous mutation in the target gene was confirmed with WB analysis. The whole cell extracts (HEK293FT WT and two knockout cell clones) were loaded to Laemmli electrophoresis in 10% SDS-PAAG and transferred on a nitrocellulose membrane (TransBlot Turbo, BIO-RAD, Hercules, CA, USA) using the semidry WB method and were incubated overnight with rabbit antibody to Tdp2 (99279 Rabbit Abcam) and β-tubulin (Abcam6046). Blots were then probed with horseradish peroxidase-coupled goat anti-rabbit antibody (1:15,000, Biotechnological Laboratory, ICBFM SB RAS, Novosibirsk, Russia) and next, we detected immunoreactivity by chemiluminescence (Pierce ECL Western blotting Substrate, Thermo Scientific, Waltham, MA, USA). Thus, we showed the absence of Tdp2 in the Tdp2-knockout cells.

### 2.4. Molecular Modeling

Crystallographic data of Tdp2 (PDB [47] code 4GYZ [48], chain B) were downloaded from the noncommercial protein data bank. Next, we prepared the geometrical parameters of Tdp2 for the calculation by adding and minimizing the hydrogen atoms, restoring the bond multiplicities and side chains of amino acids, and removing the water molecules and excess low molecular weight compounds [49]. We performed molecular docking by an induced fit docking (or flexible) protocol. The conditions of MD were as follows: flexible protein and ligand, grid size 15 Å, and a.a. within a radius of 5 Å additionally optimized considering the effect of the ligand on the side chains. We set ten docking positions. We ranked the results of the docking based on the evaluation of the scoring functions consisting of various energy terms, the docking score is the main score of docking results, the e-model is the energy of ligand clustering in the binding site, taking into account clash interaction penalties, and the IFD score is the energy of the ligand–protein complex taking into account the term of the Coulomb interaction.

## 3. Results

### 3.1. Chemistry

The compounds **11a**–**h** were synthesized by a technique described previously [44]. Thiosemicarbazones **10a**–**h** were obtained (Figure 1) using the reaction of aldehydes with thiosemicarbazide in ethanol. The formed precipitate was filtered off, then washed with water, and air dried. We obtained thiosemicarbazones **10a**–**h** with yields from 60% to 86%.

We obtained *R*-(+)-usnic acid and *S*-(−)-usnic acid by extraction from a mixture of the lichen genus *Cladonia stellaris* and *Usnea* and used the extract as a starting material [50]. The synthesis of the bromo-substituted derivatives (+)-**9** and (−)**-9** was performed by reaction of the corresponding enantiomer of usnic acid with bromine in dioxane [51]. The hydrazonothiazoles (+)-**11a**–**h** and (−)-**11a**–**h** were synthesized by the heating of the bromo-substituted derivatives **9** with thiosemicarbazones **10a**–**h** in methanol [44]. We filtered off the precipitate, dissolved it in methylene chloride, washed the precipitate with sodium bicarbonate solution, and evaporated the organic phase (Figure 1). Usnic acid derivatives (+)**-11a**–**h** and (−)**-11a**–**h** were obtained with yields from 20% to 97%.

### 3.2. Effect of Compounds on the Activity of Purified Tdp2

To evaluate the effect of the synthesized compounds on tyrosyl-DNA phosphodiesterase 2 (Tdp2) activity, we carried out the reaction of tyrosine cleavage from the 5′-end of the oligonucleotide. A fluorescent label at the 3′-end makes it possible to monitor the mobility of the initial oligonucleotide and the cleavage products in the acrylamide gel. Eight enantiomer pairs were tested, five of them had mild effect on Tdp2 activity at a concentration of 50 μM (the residual activity of the enzyme was not less than 60%). Among the three pairs remaining, **11a**,**d**,**h**, (−)-enantiomers inhibited Tdp2 in the micromolar concentration range while their (+)-counterparts at 50 μM concentration had little effect on enzyme activity. IC_50_ values (the concentration of a compound that gives half-maximal inhibition) for active (−)-derivatives are shown in Table 1.

### 3.3. Cytotoxicity Assay

Since Tdp inhibitors are intended to be used in therapeutic cocktails, non-toxic concentrations of the compounds are required so as not to exacerbate the already serious side effects of therapy. We studied the intrinsic cytotoxicity of the compounds, (−)-**11a**,**d**,**h**, that inhibited Tdp2 on HEK293FT wild-type cells and Tdp2 knockout (Tdp2^−/−^) cells (Figure 3). Tdp2 knockout HEK293FT cells were obtained using the CRISPR/Cas9 method. At 10 μM concentration, the cell survival was not less than 80% for both types of cells. This concentration was chosen for further studies of the ability of the compounds to sensitize the effect of the Top2 inhibitor etoposide.

We studied the effect of Tdp2 inhibitors on the cytotoxic effect of etoposide. The CC_50_ for etoposide was measured in the presence and absence of compounds. The sensitivity to etoposide in Tdp2 knockout cells was close to wild-type cells (Figure 4a). The cytotoxicity of etoposide increased with the addition of compounds, namely inhibitors of Tdp2 (Figure 4b–d). The significance of the differences between the metabolic activity of WT and Tdp2^−/−^ cells was assessed using the Mann–Whitney test. The points corresponding to etoposide concentrations of 0.4; 1.6; and 6 μM were compared in pairs. The maximum point of 25 μM was not considered due to too low values of the metabolic activity of the cells. For compound (−)-**11d**, the differences were significant at all three points, for (−)**11h** at points of 1.6 and 6 μM and for compound (−)-**11a** at 1.6 μM.

### 3.4. Molecular Modeling

For the elucidation of the influence of stereoisomeric usnic acid fragments on the Tdp2 inhibitory activity, the derivatives (+)-**11a**–**h** and (−)-**11a**–**h** were docked into the binding site. Due to the significant size of the molecules as well as their conformational flexibility, a number of the significant differently docking poses were obtained. Nevertheless, some poses with the specific binding pattern were found only for the (−)-enantiomers. Thus, the usnic acid fragment seemed to be a bidentate ligand of the magnesium ion located in the active site and play a key role in the Tdp2 activity [52]. The Mg^2+^ binding could be realized by C^1^=O and either C^9^-O or C^11^=O oxygen atoms of the usnic acid fragment (see the atomic numeration in Figure 5). At the same time, the usnic acid core interacts with the lipophilic flat surface formed by Leu-315 and Trp-307. The angular C-15 methyl group could be oriented to the solvent or to the small lipophilic cavity formed by Leu-134, Tyr-188, Met-214, and His-236. That orientation seems to be more preferable for both (−)- and (+)-stereoisomers. It is interesting that it is only for the (−)-enantiomers that this orientation could lead to the location of the extended substituent containing thiazole fragment in the corresponding elongated cavity (formed by Trp-307, Tyr-321, Arg-241, and His-323). A number of stacking interactions could be formed, i.e., between Trp-307, Tyr-321, and thiazole or usnic acid fragments or with aromatic systems of the substituents R in the hydrazone moiety (Figure 6). Binding of the (−)-usnic acid derivatives to the enzyme could lead to the blocking of the catalytic site containing the key amino acids such as Arg-272 and Asn-274 and a magnesium cation. The thiazole-containing tail could bind several of the amino acid residues responsible for binding to the damaged 5′-end of DNA, including His-236, Ser-239, Arg-241, Arg-276, Leu-134, Trp-307, and Leu-315, thus eliminating the ability of the protein to bind to the substrate in the catalytic site [48,53]. In summary, only the (−)-enantiomers seem to be able to bind the catalytic site and the pocket simultaneously, hindering the binding and orientation of the damaged DNA that could result in their Tdp2 inhibitory activity in contrast to (+)-isomers.

For the inactive derivatives of (−)-usnic acid, in particular compounds (−)**-11b**–**g**, less efficient binding with the extended cavity was found. Here, we considered only poses with the same pattern that explained the difference in activity for enantiomeric (−)- and (+)-usnic acid derivatives. In particular, only poses with the usnic acid core binding magnesium ion and the heterocycle tail binding the cavity Arg-241, Trp-307, Tyr-321, and His-323 were taken into account. In this context, in general, the inactive compounds (−)-**11b**–**g** demonstrated higher docking scores, being up to −8.0 kcal/mol, while the active compounds (−)-**11a**,**d**,**h** were found to bind the protein with the better docking scores being about −8.5 kcal/mol (Appendix A). For the inactive thiophene derivatives (−)-**11b**,**c**, the usnic acid fragment bound the magnesium ion as well but two substituted thiophene fragments interacted with the protein less effectively than the pyrrole fragment of (−)-**11a**. In particular, for (−)-**11b**, the only one corresponding pose was obtained. Considering the diphenyl derivatives, (−)-**11f**,**g**, the substituent seems to be too extended with a bad orientation of aromatic fragments. Thus, no corresponding poses were found at all for the (−)-**11f** and the only one was obtained for the (−)**-11g**. Comparing indole derivatives (−)-**11d** and (−)-**11e** for the inactive compound (−)-**11e**, the same poses were found as for the active compound (−)-**11d** but the (−)-**11e** was docked, with the worse docking score being −7.524 compared to −8.613 for the (−)-**11d** that could be evidence of the worse binding resulting in the loss of the activity.

## 4. Discussion

To date, hydrazinothiazoles based on usnic acid are the most effective known inhibitors of Tdp1 and this enzyme is not sensitive to the position of the angular methyl group of the usnic acid core; the IC_50_ values for both enantiomers are within the same order of magnitude [44,54,55].

In this work, we tested 8 pairs of enantiomers of hydrazinothiazoles of usnic acid as Tdp2 inhibitor; among them, 13 compounds had a low inhibitory effect on Tdp2 activity at a concentration of 50 μM. Three derivatives of (−)-usnic acid inhibit the enzyme in the micromolar concentration range: (−)-**11a**,**d**,**h**.

In the literature on the activity of usnic acid and its derivatives, there are very few works on the study of both enantiomers of compounds (since only the (+)-enantiomer is commercially available). Data on the activity of (+)- and (−)-usnic acids from these few works indicate that the difference in activity is common, but, as a rule, it is insignificant [42,43,56]. The cytotoxic effect of usnic acid and its derivatives usually does not depend on the structure of the asymmetric center if the cytotoxicity is associated with the triketone fragment of the C ring of usnic acid [42], and therefore, in those processes where this mechanism is the main one, the difference in the action of the usnic acid enantiomers and its derivatives is insignificant. A significant influence of the position of the angular methyl group is observed in cases where specific binding of usnic acid to a substrate occurs, for example, in proteins and enzymes. This is noted in publications where a usnic acid–enzyme interaction was observed [42] as well as in works on the antiviral and sensitizing effects of usnic acid and its derivatives [43]. However, there are no data on differences in effects at the level of molecular interactions in the literature.

The difference in the inhibitory action on Tdp1 of enantiomeric pairs of the compounds studied in this work was not significant. Previously, we showed [55] that the methyl group attached to an asymmetric carbon atom of the usnic moiety in both isomers forms hydrophobic contacts with the same amino acid residues of Tdp1 (the side chains of the Tyr204 and the deprotonated His263 residues) and the inhibitor molecules themselves are oriented in the active center secularly relative to each other.

However, the pairs of (+)- and (−)-usnic acid derivatives **11a**–**h** showed a difference in Tdp2 inhibition: (+)-isomers did not affect the activity of the enzyme at concentrations up to 50 μM, while the IC_50_ values for some (−)-isomers were 6–9 μM. Interestingly, according to the docking experiment, the location of the heterocyclic fragment in that cavity does not let the (+)-usnic acid be oriented similarly to its (−)-stereoisomer. On the other hand, the location of the (+)-usnic fragments in a similar way to the (−)-isomer does not let the extended heterocyclic chain bind the cavity. It seems that proper binding of the extended substituent and the preferable location of the usnic acid into the active site with simultaneous magnesium ion binding are significant for the demonstrated activity.

Thus, we have shown that (+)-isomers are selective inhibitors of Tdp1 while (−)-usnic acid derivatives, (−)-**11a**,**d**,**h**, suppress the activity of both enzymes. Such inhibitors are relevant due to the recently discovered ability of Tdp1 and Tdp2 to take on each other’s functions [24,29,30,31]. Dual Tdp1/2 inhibitors can be used to increase the efficacy of a large range of clinically important anticancer drugs (Top1/2 inhibitors) and to overcome resistance to them in various types of tumors. It should be noted that significant differences in the structure of the active centers of the Tdp1 and Tdp2 enzymes make this task extremely nontrivial. Therefore, in this work, for the first time, we measured and confirmed this by docking the difference in binding to the target (Tdp2) of different enantiomers of usnic acid derivatives.

Triple inhibitors of Tdp1/2, which also suppress the activity of Top1, were created by the group of Yves Pommier [57,58]. These inhibitors are not highly effective against purified Tdp1 and Tdp2 enzymes (ED_50_ for Tdp1/2 ranges from 1 to 40 μM) but suppress the growth of tumor cells in the nanomolar concentration range. Activity against Top1 was comparable to that of camptothecin. The authors explain this by the presence of a synergistic effect when two or three targets are suppressed at once. But the in vivo experiments with these compounds were not conducted and there are no data on their toxicity to animals.

Our team previously discovered dual inhibitors of Tdp1/2 among deoxycholic acid derivatives [41] and thioether derivatives of usnic acid [40]. These compounds were non-toxic or slightly toxic to the cells, in contrast to the isoquinolines of the Pommier’s group, but have a mild inhibitory effect against Tdp2 (in submillimolar concentrations) and were quite effective against Tdp1 (in micromolar and submicromolar concentrations). The (−)-usnic acid derivatives, (−)-**11a**,**d**,**h**, studied in the present work inhibit both enzymes much more efficiently (in the nanomolar range against Tdp1 and in the micromolar range against Tdp2) and can be used as a platform for designing dual Tdp1/2 inhibitors.

When studying the effect of compounds on the cytotoxic effect of etoposide, we expected that the compounds would have a sensitizing effect on wild-type cells while they would not have any effect on Tdp2 knockout cells due to the lack of a target. Another possibility was to see no effect on HEK293FT cells, as was the case with combinations of topotecan and Tdp1 inhibitors on related HEK293A cells [59,60]. For wild-type cells, we saw some protective effects of the compounds: an increase in CC_50_ values from 2.4 µM to 3 µM and higher (compare the black plots in Figure 4a and Figure 4b–d, respectively). As for Tdp2^−/−^ cells, we observed a two-fold decrease in CC_50_ values for an etoposide in the presence of usnic acid derivatives, specifically (−)-**11a**,**d**,**h** (red graphs in Figure 4). At the same time, in the presence of compounds, the cytotoxicity of etoposide significantly increased for Tdp2^−/−^ cells compared to HEK293FT WT (compare black and red plots in Figure 4b–d). We hypothesize that the additional toxicity of etoposide to Tdp2^−/−^ cells in the presence of the compounds is due to the effective suppression of Tdp1 in addition to the absence of Tdp2. As was mentioned above, usnic acid hydrazinothiazole derivatives are the most effective inhibitors of Tdp1 known to date and act in the nanomolar concentration range. Specific inhibitors tend to cause fewer side effects than molecules that have multiple targets. But, since the functions of Tdp1 and Tdp2 overlap, the suppression of the activity of both enzymes at once will lead to a higher efficiency of therapy. At the same time, when choosing the dose of the drug, the levels of both enzyme expressions in individual patients should be taken into account. More deep research in the future is required to understand the Tdp1 and Tdp2 interplay under the action of effective dual and selective inhibitors.

The overexpression of Tdp1 is observed in such types of cancer as non-small-cell lung cancer, colorectal cancer, breast cancer, and some rhabdomyosarcomas (reviewed in [22]). Also, it is known that Tdp2 is overexpressed in non-small cell lung carcinoma cell lines [61]. We want to highlight that given the use of topoisomerase poisons in cancer therapy, and to a particularly high level in some cancers, the overlapping of Tdp1 and Tdp2 functions in our results suggest that Tdp1/Tdp2 could be valid targets for the search for effective Tdp1/Tdp2 inhibitors as drug prototypes to improve the effectiveness of treatment and to overcome tumor resistance to topoisomerase poisons.

## 5. Conclusions

In the present work, we have shown that hydrazonothiazole derivatives of usnic acid with previously discovered inhibitory properties against Tdp1 are also capable of inhibiting Tdp2. We measured this inhibition for the first time and confirmed it by docking the difference in binding to Tdp2 of different enantiomers of usnic acid derivatives. Both (+)- and (−)-enantiomers of compounds act equally effectively against Tdp1 and only some (−)-enantiomers are active against Tdp2. The compounds protect HEK293FT WT cells from the cytotoxic effect of etoposide but potentiate it against Tdp2^−/−^ cells. Presumably, the sensitizing effect of the compounds in the absence of Tdp2 is associated with the effective inhibition of Tdp1, which could take over the functions of Tdp2. The oncogenic role of these two enzymes still needs more research but we believe that, undoubtedly, our discovery of Tdp1/Tdp2 inhibitors can make a good contribution to the development of potential drug cocktails for the treatment of various types of cancer in combination with topoisomerase poisons.

## Data Availability

Data are contained within the article or Appendix A.

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
