# Peer review of "Usnic Acid Derivatives Inhibit DNA Repair Enzymes Tyrosyl-DNA Phosphodiesterases 1 and 2 and Act as Potential Anticancer Agents"

_genes, 2023, doi:10.3390/genes14101931_

Round 1
Reviewer 1 Report
Topoisomerases are common drug targets in cancer treatment.
Camptothecin and Etoposide have been widely used in chemotherapy. They trap Top1 and Top2 at single and double-strand breaks respectively and block cancer cell growth. T
he repair of protein-DNA adducts derived from Top1 and Top2 requires Tdp1 and Tdp2 enzymes. Hence, both enzymes are promising cancer drug targets for therapies in conjunction with camptothecin and etoposide. T
his manuscript by Zakharenko et.al, developed a series of novel TDP2 inhibitors and tested them in both biochemical and cytotoxicity assays. Overall, the work has been generally well conducted and the results are interesting to the researcher in the field. one major concern is the conclusion that the authors reached based on Figure 4b-d on "Page 7", viz. "the cytotoxicity of etoposide increased with the addition of the compounds - inhibitors of Tdp2", is not obvious based on the data presented. If any statistical analysis has been conducted to reach this conclusion, it needs to be included.
Author Response
Dear Reviewer,
We are very grateful for careful reading of our manuscript, its analysis and useful comments.
Please find below our response on your comments.
Sincerely,
Zakharenko A. and coauthors
Reviewer 1. Topoisomerases are common drug targets in cancer treatment.
Camptothecin and Etoposide have been widely used in chemotherapy. They trap Top1 and Top2 at single and double-strand breaks respectively and block cancer cell growth. The repair of protein-DNA adducts derived from Top1 and Top2 requires Tdp1 and Tdp2 enzymes. Hence, both enzymes are promising cancer drug targets for therapies in conjunction with camptothecin and etoposide. This manuscript by Zakharenko et.al, developed a series of novel TDP2 inhibitors and tested them in both biochemical and cytotoxicity assays. Overall, the work has been generally well conducted and the results are interesting to the researcher in the field.
one major concern is the conclusion that the authors reached based on Figure 4b-d on "Page 7", viz. "the cytotoxicity of etoposide increased with the addition of the compounds - inhibitors of Tdp2", is not obvious based on the data presented. If any statistical analysis has been conducted to reach this conclusion, it needs to be included.
Answer: Thank you! We have compared pairwise points according to Mann-Whitney test on Figure 4b-d and have added it to the text. Also we have expanded the Conclusion a little.
Reviewer 2 Report
Genes (Manuscript ID: genes-2567336), Comments to the Authors:
Title: DNA repair enzymes Tyrosyl-DNA phosphodiesterases 1 and 2 inhibitors based on usnic acid as potential anticancer agents
Comments
The submitted manuscript provides a detailed account of the experimental findings regarding the inhibitory effects of hydrazinothiazoles derived from usnic acid on Tdp1 and Tdp2 enzymes. While the discussion presents valuable insights into the study's outcomes, several aspects need further clarification and elaboration to enhance the overall quality and impact of the section.
I think the submitted manuscript can be accepted for publication after the authors respond to the following comments:
- The title is confusing. I think the authors should change the title to “Usnic acid derivatives inhibit DNA repair enzymes Tyrosyl-DNA phosphodiesterases 1 and 2 inhibitors and act as potential anticancer agents”
- The abstract should contain numeral values so the readers can compare the submitted results with previous findings.
- Mechanistic Insights: While the discussion briefly touches upon the mechanistic aspects of Tdp1 and Tdp2 inhibition by hydrazinothiazoles, it lacks depth in explaining the underlying molecular interactions that contribute to these inhibitory effects. The authors mention hydrophobic contacts and the orientation of inhibitor molecules but do not elaborate on how these interactions impact enzyme activity. Providing a more detailed mechanistic analysis, supported by visual aids like molecular diagrams or interaction models, would clarify the proposed binding mechanisms.
- The discussion highlighted the differences in inhibitory effects between (+)- and (-)-usnic acid derivatives on Tdp2. However, it falls short in explaining the significance of these differences and how they relate to the overall findings of the study. A comprehensive analysis of the structural features responsible for the observed variations, supported by relevant literature or additional molecular modeling data, would provide a more robust understanding of the experimental outcomes.
- While the discussion briefly mentions the potential use of dual Tdp1/2 inhibitors in cancer therapy, it lacks a thorough exploration of the broader implications and clinical relevance of the study's findings. Expanding on how the discovered selective and dual inhibitors could impact cancer treatment, especially in combination with existing anticancer drugs, would provide a more compelling argument for the significance of the research.
- The authors touch upon unexpected results related to the cytotoxic effect of etoposide in the presence of usnic acid derivatives. However, the discussion fails to thoroughly address the implications of these findings and how they contribute to the overall understanding of Tdp1 and Tdp2 interactions. A more detailed analysis of these unexpected results, along with potential mechanisms underlying the observed effects, would strengthen the discussion's impact.
Author Response
Dear Reviewer,
We are very grateful for careful reading of our manuscript, its analysis and useful comments.
Please find below our point by point responses on all your comments.
Sincerely,
Zakharenko A. and coauthors
Reviewer 2. The submitted manuscript provides a detailed account of the experimental findings regarding the inhibitory effects of hydrazinothiazoles derived from usnic acid on Tdp1 and Tdp2 enzymes. While the discussion presents valuable insights into the study's outcomes, several aspects need further clarification and elaboration to enhance the overall quality and impact of the section.
I think the submitted manuscript can be accepted for publication after the authors respond to the following comments:
The title is confusing. I think the authors should change the title to “Usnic acid derivatives inhibit DNA repair enzymes Tyrosyl-DNA phosphodiesterases 1 and 2 inhibitors and act as potential anticancer agents”
Answer: We really appreciate this suggestion. The title of the manuscript was changed according to this comment.
The abstract should contain numeral values so the readers can compare the submitted results with previous findings.
Answer: We have added the required information to the abstract.
Mechanistic Insights: While the discussion briefly touches upon the mechanistic aspects of Tdp1 and Tdp2 inhibition by hydrazinothiazoles, it lacks depth in explaining the underlying molecular interactions that contribute to these inhibitory effects. The authors mention hydrophobic contacts and the orientation of inhibitor molecules but do not elaborate on how these interactions impact enzyme activity. Providing a more detailed mechanistic analysis, supported by visual aids like molecular diagrams or interaction models, would clarify the proposed binding mechanisms.
Answer: We have added extended characterization of this interaction in the section 3.3 Molecular Modeling and also have done it in Discussion.
The discussion highlighted the differences in inhibitory effects between (+)- and (-)-usnic acid derivatives on Tdp2. However, it falls short in explaining the significance of these differences and how they relate to the overall findings of the study. A comprehensive analysis of the structural features responsible for the observed variations, supported by relevant literature or additional molecular modeling data, would provide a more robust understanding of the experimental outcomes.
Answer: We have added a few sentences to the Discussion on this topic. There is no relevant literature (targeted studies for pairs of enantiomers of usnic acid or its derivatives), so it is difficult to discuss the relationship between the observed differences and the obtained results. It probably makes sense in the future to make a number of additional modifications to the compounds to establish this relationship.
While the discussion briefly mentions the potential use of dual Tdp1/2 inhibitors in cancer therapy, it lacks a thorough exploration of the broader implications and clinical relevance of the study's findings. Expanding on how the discovered selective and dual inhibitors could impact cancer treatment, especially in combination with existing anticancer drugs, would provide a more compelling argument for the significance of the research.
Answer: Specific inhibitors tend to lead to fewer side effects than molecules that have multiple targets. On the other hand, since the functions of Tdp1 and Tdp2 overlap, the suppression of the activity of two enzymes at once will lead to a greater effectiveness of therapy. At the same time, when choosing a dose of the drug, the levels of enzyme expression in individual patients should be taken into account. Of course, this requires in-depth research, which is far beyond the scope of this work.
The authors touch upon unexpected results related to the cytotoxic effect of etoposide in the presence of usnic acid derivatives. However, the discussion fails to thoroughly address the implications of these findings and how they contribute to the overall understanding of Tdp1 and Tdp2 interactions. A more detailed analysis of these unexpected results, along with potential mechanisms underlying the observed effects, would strengthen the discussion's impact.
Answer: Thank you! We tried to expand Discussion and Conclusion.
Reviewer 3 Report
In this manuscript, eight pairs of enantiomeric analogues of usnic acid were synthesized. Their enzymatic inhibition activity against Tdp2 was evaluated. Three low micromolar inhibitors were identified. Docking studies were performed to rationalize activity for inhibiting Tdp2. A Tdp2 knock-out HEK293FT cell line was successfully generated using CRISP-Cas9. Cytotoxicity of the three active compounds was then assessed against both wide-type and Tdp2 knock-out HEK293FT cells. The experimental data are solid, but unfortunately, functional overlap of Tdp1 and Tdp2 made data-interpretation challenging and inconclusive. The long-term goal of the work here is to find dual inhibitors against both Tdp1 and Tdp2, therefore capable of minimizing resistance of clinically used DNA-targeting anticancer drugs. Therefore, the research project well matches the scope of the journal and is suitable for publication. However, it would be helpful if the authors could address the following points during their revision.
Major point:
Given the functional overlap of Tdp1 and Tdp2, as a control, it would be better to check expression of Tdp1 and Tdp2 at the protein level for one of the three inhibitors. Could inhibition or knock-out of Tdp2 loop back to cause overexpression of Tdp1? If such experiments have been done before, related literature may be cited.
Minor points:
1) In the last line of the third paragraph on page 2, there should be a space between “inhibit” and “Tdp1”.
2) In the last paragraph of page 3, what is the concentration/amount of BSA?
3) For figure 4, would a log scale for concentration be better?
4) In the second last line of the last paragraph on page 9, would “worth” be “worse”?
Author Response
Dear Reviewer,
We are very grateful for careful reading of our manuscript, its analysis and useful comments.
Please find below our point by point responses on all your comments.
Sincerely,
Zakharenko A. and coauthors
Reviewer 3. In this manuscript, eight pairs of enantiomeric analogues of usnic acid were synthesized. Their enzymatic inhibition activity against Tdp2 was evaluated. Three low micromolar inhibitors were identified. Docking studies were performed to rationalize activity for inhibiting Tdp2. A Tdp2 knock-out HEK293FT cell line was successfully generated using CRISP-Cas9. Cytotoxicity of the three active compounds was then assessed against both wide-type and Tdp2 knock-out HEK293FT cells. The experimental data are solid, but unfortunately, functional overlap of Tdp1 and Tdp2 made data-interpretation challenging and inconclusive. The long-term goal of the work here is to find dual inhibitors against both Tdp1 and Tdp2, therefore capable of minimizing resistance of clinically used DNA-targeting anticancer drugs. Therefore, the research project well matches the scope of the journal and is suitable for publication. However, it would be helpful if the authors could address the following points during their revision.
Major point:
Given the functional overlap of Tdp1 and Tdp2, as a control, it would be better to check expression of Tdp1 and Tdp2 at the protein level for one of the three inhibitors. Could inhibition or knock-out of Tdp2 loop back to cause overexpression of Tdp1? If such experiments have been done before, related literature may be cited.
Answer: We appreciate this suggestion, it could really enhance the overall quality of the work and help to contribute to the overall understanding of Tdp1 and Tdp2 interactions. But conducting such additional experiments will take time. Previously, we performed a transcriptomic analysis of wild-type and Tdp1 knockout HEK293A cells in the presence of another usnic acid derivative that effectively inhibits Tdp1, but not Tdp2. We did not find significant changes in Tdp2 expression either under the influence of Tdp1 knockout or under the influence of an inhibitor of the latter. These data are still being prepared for publication, so we cannot refer to them. The reverse situation, Tdp1 expression during Tdp2 knockout or inhibition, has not been studied. If the Reviewer insists and the Editors approve, we can perform expression analysis, but we will need more time, at least until the end of October.
Minor points:
1) In the last line of the third paragraph on page 2, there should be a space between “inhibit” and “Tdp1”.
Answer: Thanks, it was corrected
2) In the last paragraph of page 3, what is the concentration/amount of BSA?
Answer: corrected
3) For figure 4, would a log scale for concentration be better?
Answer: corrected
4) In the second last line of the last paragraph on page 9, would “worth” be “worse”?
Answer: Sure, sorry, it was corrected.
Reviewer 4 Report
In this paper the authors wanted to show the effect of their TDP2 inhibitors on HEK cell line. Even the concept is interesting a lot of supplemental experiments must be done to complete this study.
Fig. 1 and Fig. 2 should be moved to the supplemental section, there were not pertinent in my opinion in the main text. How the IC50 displayed were calculated? By the authors or found in the literature?
Fig. 3 Authors should add significant test. How the inhibitor of TDP2 affects more the KO cell lines compared to WT? What about the specificity of these inhibitors? Authors must display these results.
Author Response
Dear Reviewer,
We are very grateful for careful reading of our manuscript, its analysis and useful comments.
Please find below our point by point responses on all your comments.
Sincerely,
Zakharenko A. and coauthors
Reviewer 4. In this paper the authors wanted to show the effect of their TDP2 inhibitors on HEK cell line.
Even the concept is interesting a lot of supplemental experiments must be done to complete this study.
Answer: Thank you for your interest in our work! Really, in this work we found for the first time effective Tdp2 inhibitors among usnic acid derivatives, we measured and confirmed by docking the difference in binding to the target enzyme of different enantiomers of this natural compound (usnic acid). We kindly ask Editors and Reviewers about two months for cell experiments that could enhance the quality of our work according to Reviewers’ comments. Please see the details below in our point by point responses on your comments.
Tdp1 and Tdp2 are often upregulated in lung cancer. The authors in order to strengthen their argument around IC50, should also run dose dependent studies on some lung cancer cell lines to determine the true nature of the IC50 and whether the compounds 11a, d and h are indeed universal in their activity across multiple cell lines.
Answer: We also thank the referee for a good idea. We have cancerous and non-cancerous lung cells in our institute's collection, and such a study would really be useful. We are ready to perform these experiments if the editors approve, but we need more time.
Fig. 1 and Fig. 2 should be moved to the supplemental section, there were not pertinent in my opinion in the main text. How the IC50 displayed were calculated? By the authors or found in the literature?
Answer: In the introduction we tried to give an idea of the development of the topic. And since there are specialists of various profiles among the readers, we believe that such a presentation will be interesting, especially for chemists. Some of the IC50 values were found in the literature and some of them were measured in our previous studies according to the References.
Fig. 3 Authors should add significant test. How the inhibitor of TDP2 affects more the KO cell lines compared to WT? What about the specificity of these inhibitors? Authors must display these results.
Answer: Thank you! We have compared pairwise points according to Mann-Whitney test on Figure 4b-d and have added it to the text. Since no one else has studied these compounds, we only know about their specificity that compounds of this type effectively inhibit Tdp1 (doi:10.3390/molecules24203711), while inhibition of Tdp2 depends dramatically on the substituent in the hydrazonothiazole fragment (present work).
We think that additional cell experiments could enhance the overall quality of the work and help to contribute to the overall understanding of Tdp1 and Tdp2 interactions. But conducting such additional experiments will take time. Previously, we performed a transcriptomic analysis of wild-type and Tdp1 knockout HEK293A cells in the presence of another usnic acid derivative that effectively inhibits Tdp1, but not Tdp2. We did not find significant changes in Tdp2 expression either under the influence of Tdp1 knockout or under the influence of an inhibitor of the latter. These data are still being prepared for publication, so we cannot refer to them. The reverse situation, Tdp1 expression during Tdp2 knockout or inhibition, has not been studied. If the Reviewer insists and the Editors approve, we are planning to perform expression analysis, but we will need more time, at least until the end of October.
Reviewer 5 Report
Zakharenko et al., in the above article discusses the roles of DNA repair enzymes Tdp1 and Tdp2 in repairing DNA damage caused by various agents, including anticancer drugs. The study moreover identifies compounds derived from (-) - usnic acid that can inhibit both Tdp1 and Tdp2 which could be further leveraged to develop a Tdp1/Tdp2 dual inhibitor.
While the premise of the study is neat, the authors should address the following concerns -
1. In the introduction, the authors need to include the clinical relevance of Tdp1 and Tdp2 in cancer. For eg, the authors can highlight in which cancer type/s are Tdp1 and Tdp2 over-expressed or where resistance to therapy develops due to Tdp1 and Tdp2. They can also include the percent patient cases to justify the targets.
2. The authors should elaborate whether Tdp1 and Tdp2 regulate each other and if possible look at Tdp1 levels in Tdp2 KO cells.
3. Tdp1 and Tdp2 are often upregulated in lung cancer. The authors in order to strengthen their argument around IC50, should also run dose dependent studies on some lung cancer cell lines to determine the true nature of the IC50 and whether the compounds 11a, d and h are indeed universal in their activity across multiple cell lines.
Author Response
Dear Reviewers,
We are very grateful for careful reading of our manuscript, its analysis and useful comments.
Please find below our point by point responses on all your comments.
Sincerely,
Zakharenko A. and coauthors
Reviewer 5. Zakharenko et al., in the above article discusses the roles of DNA repair enzymes Tdp1 and Tdp2 in repairing DNA damage caused by various agents, including anticancer drugs. The study moreover identifies compounds derived from (-) - usnic acid that can inhibit both Tdp1 and Tdp2 which could be further leveraged to develop a Tdp1/Tdp2 dual inhibitor.
While the premise of the study is neat, the authors should address the following concerns -
In the introduction, the authors need to include the clinical relevance of Tdp1 and Tdp2 in cancer. For eg, the authors can highlight in which cancer type/s are Tdp1 and Tdp2 over-expressed or where resistance to therapy develops due to Tdp1 and Tdp2. They can also include the percent patient cases to justify the targets.
Answer: Thank you for the good suggestion! We have added information about Tdp1 and Tdp2 overexpression in cancer cells to Discussion.
The authors should elaborate whether Tdp1 and Tdp2 regulate each other and if possible look at Tdp1 levels in Tdp2 KO cells.
Answer: We appreciate this suggestion, it could really enhance the overall quality of the work and help to contribute to the overall understanding of Tdp1 and Tdp2 interactions. But conducting such additional experiments will take time. Previously, we performed a transcriptomic analysis of wild-type and Tdp1 knockout HEK293A cells in the presence of another usnic acid derivative that effectively inhibits Tdp1, but not Tdp2. We did not find significant changes in Tdp2 expression either under the influence of Tdp1 knockout or under the influence of an inhibitor of the latter. These data are still being prepared for publication, so we cannot refer to them. The reverse situation, Tdp1 expression during Tdp2 knockout or inhibition, has not been studied. If the Reviewer insists and the Editors approve, we can perform expression analysis, but we will need more time, at least until the end of October.
Tdp1 and Tdp2 are often upregulated in lung cancer. The authors in order to strengthen their argument around IC50, should also run dose dependent studies on some lung cancer cell lines to determine the true nature of the IC50 and whether the compounds 11a, d and h are indeed universal in their activity across multiple cell lines.
Answer: We also thank the referee for this good idea. We have cancerous and non-cancerous lung cells in our institute's collection, and such a study would really be useful. We are ready to perform these experiments if the editors approve, but we need more time.
Round 2
Reviewer 1 Report
My previous concerns have been properly addressed.
Reviewer 2 Report
Genes (Manuscript ID: genes-2567336 - Revised Version), Comments to the Authors:
Title: DNA repair enzymes Tyrosyl-DNA phosphodiesterases 1 and 2 inhibitors based on usnic acid as potential anticancer agents
Comments
After reading the authors’ response to my comments, I think the revised manuscript can be accepted for publication.
Reviewer 4 Report
In my opinion, my previous comments should be done to complete this paper.